# Impact of demographic factors and HIV status on Hepatitis B vaccination adherence and completion rate among high-risk populations in Lagos State, Nigeria

Elizabeth Shoyemi, Ojore Godday Aghedo 🔟*, Collins Obi 🔟,
Chukwuebuka Peter Kelvin Uwaoma 🔟, Oluwafisayomi S. Azeez 🔟, Okikiola Oyeniyi 🔟

Research Department, Centre for Population Health Initiatives, Lagos State, Nigeria

* aghedogodday@gmail.com

## Abstract

Hepatitis-B-Virus (HBV) remains an important global public health concern, as well as a primary cause of both acute and chronic liver diseases, with over 18 million people infected by the virus in Nigeria. This study evaluated the impact of HIV status, age, and gender of high-risk populations on HBV vaccination adherence and completion rate in Lagos state, Nigeria. A retrospective study design was utilized to access the clinic data of 641 study participants who participated in the HBV program at the one-stop-shop clinic between June 2021 to June 2024. Data on age, HIV status, gender, population type, and the date of each dose of administered vaccine were recorded. To estimate the odds-ratios (ORs) and adjusted-ORs at 95% confidence intervals (CIs), bivariate/multivariate logistic regression models were utilized, and the association between age, gender, HIV status, and HBV vaccination adherence and the completion rate was examined. The study participants had a mean age of 21.7 ± 8.8 years and were composed of 510 (79.6%) males, 131 (20.4%) females; 162 (25.3%) participants living with HIV, 423 (66.0%) HIV-negative participants, 500 (78.0%) MSM, and 115 (17.9%) FSW. The overall vaccination completion rate was 45.4%, with significant variation by age group (p = 0.004). The highest completion rates were observed among participants aged 30–34 years (58.4%) and 35–39 years (57.8%), while younger participants aged 15–29 had lower adherence. Gender and HIV status were not independently associated with vaccination completion (p > 0.05). Age was the only factor significantly affecting completion (AOR: 1.92, 95% CI: 1.23, 3.03). HBV vaccination coverage in high-risk populations is low in Lagos State. This suggests that the HBV vaccination program in the state is still facing significant challenges. Creating awareness about HBV infection and strengthening the availability, accessibility, and cost effectiveness of HBV service uptake will enhance the program's success.

**Data availability statement:** The secondary data used in this study was generated from the community-based vaccination program implemented by CPHI. The data are available at the OSS clinic of CPHI, and publicly available at Figshare: https://doi.org/10.6084/m9.figshare.30998749. Also, all data related requests or inquiries can be sent to info@cphinigeria.org or to the corresponding author.

**Funding:** The author(s) received no specific funding for this work.

**Competing interests:** The authors have declared that no competing interests exist.

## Introduction

Hepatitis B Virus (HBV) remains an important global public health concern, as well as a primary cause of both acute and chronic liver diseases [1]. Late detection and poor management of HBV infection can lead to life-threatening complications like the liver cancer also called hepatocellular carcinoma (HCC), liver cirrhosis, and eventual death [2]. The transmission routes for HBV are essentially the same as those of HIV, which include contact with infected blood or body fluids, unsafe injections, exposures to contaminated sharp instruments, and mostly through mother-to-child-transmission (MTCT) during birth [1,3]. More than 250 million people lived with HBV in 2022, causing more than 1 million deaths in the same year, a decline from over 1.2 million deaths recorded in 2015, according to the World Health Organization (WHO) [1]. However, there are safe, readily accessible, and effective vaccines for preventing HBV in both children and adults [2,4].

The prevalence of HBV in Sub-Saharan Africa is quite high due to behavioral risk factors, and particularly as high as 11% − 13% in Nigeria [5,6]. Globally, the widespread administration of the safe and effective HBV vaccine has proven to be resourceful in preventing new HBV infections [4]. While global vaccination programs aim to achieve over 90% coverage, many sub-Saharan regions, including Nigeria, remain significantly behind, posing a persistent public health risk [7]. Hepatitis B vaccination is widely available and administered to infants routinely across many health facilities in Nigeria, both private and public health sector. It is offered free of charge in public primary healthcare centers (PHCs) as part of the pentavalent vaccine schedule under Nigeria's National Program on Immunization (NPI). However, outside routine infant immunization, fees are charged for adolescents and adults receiving HBV vaccination, especially in private healthcare facilities. The costs of HBV vaccination differ in the private health sector depending on the categories of the healthcare provider and the location. This potentially constitutes a financial barrier to vaccine uptake in Nigeria [5,8]. All HBV vaccination in Nigeria is guided by the National Policy on Viral Hepatitis, the NPI guidelines, and alignment with the recommendations based on WHO hepatitis elimination targets. The vaccines are administered in three doses at 0-1-6 schedule (baseline, one month and six months) to be considered full or complete vaccination, but there have been reports of suboptimal completion rates, especially amongst the key populations in Nigeria and Sub-Sahara Africa [2,4,8].

There are over 18 million people living with HBV in Nigeria, making it a high burden country for HBV [5,6,9]. Suboptimal knowledge of HBV prevention and transmission routes in more than 80% of hospital janitorial staff in Nigeria, has been reported in a cross-sectional study involving tertiary hospital cleaners in Nigeria [10]. Low level or absolute lack of awareness of hepatitis B, and suboptimal knowledge of its prevention and transmission routes, were identified in the study, as a reason for poor participation in vaccination programs by individuals [10]. Previous studies have associated level of knowledge and status of HBV vaccination with sociodemographic factors like age and marital status [11–13].

Lagos State is the economic hub of Nigeria. However, high-risk groups, like people living with HIV (PLHIV), and key populations like female sex workers (FSW) and men

who have sex with men (MSM) are extremely affected by Hepatitis B virus, and they constitute substantial healthcare challenges faced in the state [14–16], forming a basis for this study.

The immuno-compromised state associated with HIV increases the vulnerability and susceptibility of PLHIV to infections like Hepatitis B virus [14,17]. Barriers to vaccination, competing healthcare priorities, absolute lack of or low awareness levels, and healthcare services access limitations are additional challenges PLHIV encounter [17].

A long-term protection against HBV largely depends on the completion rate of HBV vaccination series, which are often influenced by demographic factors like age, gender, and HIV status [4,8]. Younger persons who believe that they are less vulnerable to Hepatitis B, are less likely to complete the HBV vaccination regimen compared to older individuals with the perception of higher risk of HBV infection [18]. Gender and culture related factors have been reported to affect healthcare access and vaccine uptake [19]. Given that PLHIV face exclusive challenges like side effects of their medication and challenging healthcare requirements, HIV status is considered a key determinant of vaccine completion [20].

The impact of demographic variables on vaccine uptake has been studied in the past, but only few studies have been conducted on how the demographic variables directly impact HBV vaccine completion rate in high-risk populations, especially in Lagos state, Nigeria. The development of data-driven strategies and interventions aimed at both improving vaccination coverage and shrinking the prevalence of HBV, is predicated on the knowledge of the impact of factors such as age, gender, and HIV status on HBV vaccine uptake, especially among high-risk populations [21,22]. The aim of this study is to assess the influence of selected demographic factors on the HBV vaccine completion rate among high-risk populations in Lagos State, Nigeria. Future vaccination campaigns targeted at improving vaccine completion rates in vulnerable populations, will benefit from the findings of this study as a guide. The ambitious global target of exterminating HBV by 2030 is achievable if public health strategies and interventions directly improve vaccination coverage and reduce the prevalence of Hepatitis B virus [21].

### Scope and limitation of this study

This study was conducted with only the clients who voluntarily accessed HBV vaccination services at the One-Stop-Shop (OSS) Clinic of Centre for Population Health Initiatives (CPHI) in Lagos State. Therefore, the findings may not be generalizable to other regions or healthcare settings across Nigeria or sub-Saharan Africa. Being a retrospective study, it relied on secondary data collected from clinic records, which may limit the researcher from capturing other sociodemographic, cultural, or behavioral factors influencing vaccination adherence and completion. The number of observed subjects was disproportionate across the gender and population types.

## Materials and Methods

### Study location

Lagos State is arguably the most populous state in Nigeria. According to the 2021 edition of 'Spotlight on Lagos Statistics,' by Lagos State Bureau of Statistics, the commercial hub boasts of an estimated population figure of more than 25 million inhabitants, which is higher than the official 2006 census figure of 9,013,534 inhabitants. Lagos State has 20 local government areas (LGAs), and its annual growth rate is between 2.5% and 3.5%, with a population density of more than 4,139 persons per square kilometer [5]. Research activities took place at the OSS Clinic in Yaba, Lagos State, managed by CPHI, which has recorded a high volume of high-risk populations accessing health services. CPHI is a non-profit, non-governmental organization that conducts cutting-edge research activities and implements various funded public health projects with a special focus on high-risk populations. The study incorporated participants from all the 20 LGAs of the state who accessed health services at the facility.

### Study design

The study employed retrospective design by accessing participants' clinic data from June 1st, 2021, to June 30th, 2024. The vaccination program started in 2021, so no routinely collected programmatic data prior to 2021 was available for

inclusion in this study. Complete secondary data was accessed on January 6th, 2025, for the purpose of this study. The documentation tools (such as the registers, HBV enrolment forms) were reviewed for participants who started the vaccine regimen within the period of interest. Afterwards the adherence level and completion rate for the recommended doses of Hepatitis B vaccine was examined. This study design is suitable for evaluating the rate of adherence and completion of HBV vaccine, and how it is influenced by sociodemographic factors like age, gender and HIV status. A validation process such as double-checking records for accuracy and consistency was employed to mitigate potential biases associated with the study design. Also, authors did not have access to information that could identify individual participants during or after data collection process.

## Sample size and study population

The study involved 641 participants who initiated their doses of the HBV vaccine at the OSS Clinic within the cohort period of interest, across all age groups. The standard vaccination schedule of 0-1-6 was adopted. This sample size of participants was determined based on prior studies on HBV vaccine adherence within high-risk populations by LaMori et al., [8], highlighting a minimum of 500 participants being required to detect the meaningful associations between demographic factors and vaccine adherence. The study population was constituted by Men who have Sex with Men (MSM), Female Sex Workers (FSW) and partners of MSM and FSW representing the general populations.

## Inclusion and exclusion criteria

Participants who received the first dose of the HBV vaccine at least six months prior to the commencement of the study at the CPHI's OSS clinic were included. This is to ensure accurate assessment of adherence and completion rate of all three doses of the vaccines. Participants must have complete record of age, sex, dates of each dose of vaccine received, population type, and vaccination outcome. Participants with incomplete records, or received vaccinations outside the defined timeframe, and participants with inconsistent or conflicting data were excluded from the study.

## Ethics Statement

The Institutional Review Board (IRB) of the Nigerian Institute for Medical Research (NIMR) issued the ethical clearance for this study. Approval number: IRB/24/083. Consent was not required as the study team had no direct contact with the subjects. Confidentiality and privacy of data was maintained throughout the research processes, and personally identifying information was not collected, unique IDs were used instead. Both national and institutional ethical guidelines were strictly observed in the conduct of this research.

## Data collection process

Study data of all participants who received HBV vaccinations from the CPHI's OSS clinic were retrieved from HBV vaccination registers and enrollment forms. Data was entered into a specially formatted Microsoft Excel template and revalidated by a study staff for data accuracy and completeness. Data cleaning and coding were carried out using Microsoft Excel spreadsheet (version 2024) and then imported into IBM SPSS Statistics (version 26) for further statistical analysis. To estimate the odds ratios (ORs) and adjusted ORs at 95% confidence intervals (CIs), bivariate and multivariate logistic regression analysis was deployed, and the association between age, gender, HIV status and HBV vaccination adherence and completion rate was assessed.

## Operational definition of HBV vaccination outcomes

- **Completed:** All persons who received all the expected doses of the HBV vaccine in line with the standard vaccination schedule of 3 doses within 6 months, are considered to have "completed" the HBV vaccination regimen.

- **Not completed:** All persons who did not receive all the expected 3 doses of the HBV, are considered to have "Not completed" the HBV vaccination regimen, regardless the number of doses they have taken already.

- **Adherent:** All persons who received all the expected 3 doses of the HBV vaccine by following the stipulated vaccination schedules, are classified as "Adherent". This means, they received the vaccine doses at the proper intervals or within acceptable delays between doses, typically 14 days.

- **Not adherent:** All individuals who failed to receive all the expected 3 doses of the HBV vaccine within the recommended vaccination schedules, are classified as "Not adherent".

### Data variables and analysis

The variables collected were categorical variables and continuous variables, such as Gender, Age group, HIV status, Vaccination status, Dates of First, Second, and Third doses. All data analyses were done using Microsoft Excel pivot tables and SPSS version 26. Descriptive statistics in the form of frequency and percentages were completed for categorical variables.

The completion rate was calculated as:

$$\frac{\textit{Number of participants who completed the vaccine series within specified time}}{\textit{Number of participants who initiated the first dose of vaccine}} X100$$

Adherence was measured by noting the participants who took their second and third doses of the vaccine within the stipulated time including a window period of 14 days. The association between demographic factors, adherence and vaccine completion rate was tested using Chi-square, at a statistically significant level of less than 0.05.

### Results

A total of 641 persons were vaccinated within the period under review at CPHI and they met the inclusion criteria for this study. See Table 1 below for participants' demographic features.

The mean age of study participants was 21.7±8.77 years, with most participants within the age groups of 20–24 years and 25–29 years old. Out of the eligible participants, 79.6% (n=510) were males while 20.4% (n=131) were females. 25.3% (n=162) of the participants were persons living with HIV, 66.0% (n=423) were HIV negative and 8.7% (n=56) did not disclose their HIV status. Most participants were MSM, 78.0% (n=500), followed by FSW, 17.9% (n=115) and then the general population, 4.1% (n=26). The HBV vaccination status across key demographic characteristics is presented in Table 2.

There was a significant drop out in the number of persons who followed through the vaccination series. Out of the 641 participants, 59.8% (n=383) and 45.4% (n=291) took the second and third doses of the HBV vaccine respectively. Higher uptake of HBV vaccination was observed among male subjects, those within the age group of 25–29 years, the HIV-negative individuals, and participants classified as MSM. The vaccine coverage was significantly lower in older age groups. Table 3 compares the vaccination completion rate across the various features, including the statistical significance level.

The male participants had 44.7% vaccine adherence and completion rate, which is slightly lower than 48.1% recorded by the female participants. The completion rate was highest amongst participants in the 30–34 years age group, at 58.4%, followed by 35–39 years at 57.8%. Both HIV positive and HIV negative participants recorded the same level of adherence and completion rate (46.3%), while those with undisclosed HIV status had the lowest completion rate of 35.7%. The MSM group had a slightly higher adherence and completion rate (44.8%) than the FSW group (44.3%). Participants in the General Population (GP) group had the highest vaccine adherence and completion rate (61.5%). However, no statistically significant association existed amongst the gender, HIV status, and 'population types' categories (p > 0.05), apart from the age categories (p = 0.004).

**Table 1. Demographic characteristics of the participants.**

| Characteristics | Gender | | |
|---|---|---|---|
| | Male n (%) | Female n (%) | Total n (%) |
| **Age(years)** | | | |
| 15-19 | 14(2.7) | 4(3.1) | 18(2.8) |
| 20-24 | 112(22.0) | 38(29.0) | 150(23.4) |
| 25-29 | 208(40.8) | 55(42.0) | 263(41.0) |
| 30-34 | 122(23.9) | 15(11.5) | 137(21.4) |
| 35-39 | 38(7.5) | 7(5.3) | 45(7.0) |
| 40-44 | 13(2.5) | 5(3.8) | 18(2.8) |
| 45-49 | 2(0.4) | 4(3.1) | 6(0.9) |
| 50-54 | 1(0.2) | 1(0.1) | 2(0.3) |
| 55-59 | 0(0.0) | 1(0.1) | 1(0.2) |
| ≥ 60 | 0(0.0) | 1(0.1) | 1(0.2) |
| **HIV Status** | | | |
| Positive | 142(27.8) | 95(72.5) | 162(25.3) |
| Negative | 328(64.3) | 16(12.2) | 423(66.0) |
| Not Disclosed | 40(7.8) | 20(15.3 | 56(8.7) |
| **Population Type** | | | |
| MSM | 500(98.0) | 0(0.0) | 500(78.0) |
| FSW | 0(0.0) | 114(87.9) | 114(17.8) |
| GP | 10(2.0) | 17(12.1) | 27(4.2) |
| **Total** | **510(100.0)** | **131(100.0)** | **641(100.0)** |

## Discussion

This study's findings provide essential insights into the factors influencing the completion rate of the Hepatitis B vaccine among high-risk populations in Lagos State, Nigeria. The vaccine completion rate is 45.4%, which is lower than the HBV vaccination coverage achieved in Germany and the global target of 95% by 2023 [7]. The implication of this finding is that there are either existing or potential challenges directly hindering optimal HBV vaccination amongst high-risk populations. The integration of HBV vaccination into routine clinical services and health programs will address some of the existing and potential challenges.

### Effect of age on vaccination adherence and completion rate

Previous studies in Northeastern Ethiopia [6] in China [23] and in Sierra Leone [24], reported that age has statistically significant impact on Hepatitis B vaccination adherence and completion rate. This is corroborated by findings from this study, which found that younger individual between the ages of 15–29 years, and older individuals between the ages of 40–60 years, had a lower vaccine completion rate compared to those between the ages of 30–39 years ($x^2$=2.321, df=9, p-value=0.004). Another study proved that younger persons are subject to vaccination barriers, which can include health services access restrictions and inaccurate perception of lower risk of infection [18,25]. Despite these barriers, a study on adult HBV vaccination coverage conducted in China, found that vaccine completion rate among the older population was lower than in younger populations [23]. But a study conducted in Guinea and Liberia, did not report any statistically significant relationship between age and HBV vaccine completion rate in both countries, with p-values of 0.104 and 0.419 respectively [24].

Meanwhile, at a p-value less than 0.05, participants within the age group of 30–39 years in this study were twice more likely to complete the vaccination series, when compared to other age groups (AOR: 1.92, 95% CI:1.23, 3.03). Higher

**Table 2. Hepatitis-B vaccination status among the subjects.**

| Characteristics | Vaccination | | |
|---|---|---|---|
| | First Dose *n* (%) | Second Dose n (%) | Third Dose n (%) |
| **Gender** | | | |
| Male | 510(79.6) | 299(78.1) | 228(78.4) |
| Female | 131(20.4) | 84(21.9) | 63(21.6) |
| **Total** | **641(100)** | **383(100)** | **291(100)** |
| **Age (Years)** | | | |
| 15-19 | 18(2.8) | 9(2.3) | 6(2.1) |
| 20-24 | 148(23.4) | 85(22.2) | 63(21.6) |
| 25-29 | 263(41.0) | 150(39.2) | 108(37.1) |
| 30-34 | 137(21.4) | 94(24.5) | 80(27.5) |
| 35-39 | 45(7.0) | 29(7.6) | 26(8.9) |
| 40-44 | 18(2.8) | 9(2.3) | 4(1.4) |
| 45-49 | 6(0.9) | 4(1) | 2(0.7) |
| 50-54 | 2(0.3) | 1(0.3) | 0(0) |
| 55-59 | 1(0.2) | 1(0.3) | 1(0.3) |
| ≥60 | 1(0.2) | 1(0.3) | 1(0.3) |
| **Total** | **641(100)** | **383(100)** | **291(100)** |
| **HIV Status** | | | |
| Positive | 162(25.3) | 94(24.5) | 75(25.8) |
| Negative | 423(66.0) | 265(69.2) | 196(67.4) |
| Not Disclosed | 56(8.7) | 24(6.3) | 20(6.8) |
| **Total** | **641(100)** | **383(100)** | **291(100)** |
| **Population Type** | | | |
| MSM | 500(78.0) | 293(76.5) | 224(77.0) |
| FSW | 114(17.8) | 71(18.5) | 50(17.2) |
| GP | 27(4.2) | 19(5.0) | 17(5.8) |
| **Total** | **641(100)** | **383(100)** | **291(100)** |

level of awareness and knowledge of the risks associated with Hepatitis B may have contributed to this difference [25]. Lower HBV vaccination completion rate in younger populations is also attributable to demographic, parental influence, socio-economic and cultural factors [21,22,25]. From general stands, migration, and non-permanent employment or living conditions appear to be more common in the younger populations, and it can pose additional obstacles to completion of HBV vaccination series [21]. In urban areas like Lagos state where fleeting accommodation arrangements are common, to effectively target this age group, public health strategies should focus on increasing awareness of the risks of Hepatitis B and the benefits of vaccination [21,22].

### Effect of gender on vaccination adherence and completion rate

From table 3 above, female participants scored vaccination completion rate of 48.1% which is higher than the overall 45.4% and the 44.7% for male participants (AOR: 0.8, 95% CI: 0.6, 1.1). Females in sub-Sahara Africa have previously been reported to have higher tendency of seeking proper health-seeking behaviors like routine medical checkups, adherence to medication and vaccine uptake, than their male counterparts [26–28]. Nevertheless, no statistically significant association was observed between gender and Hepatitis B vaccine completion rate ($\chi^2 = 0.482$, df = 1, p-value = 0.488). This is consistent with the report of no gender influence on the completion of vaccination series in another study that

 

**Table 3. Completion rate by demographic characteristics.**

| Characteristics | Vaccination Status | | | | | |
|---|---|---|---|---|---|---|
| | Total | Completed n (%) | Not Completed n (%) | χ2 | df | p-Value |
| **Gender** | | | | | | |
| Male | 510(100.0) | 228(44.7) | 282(55.3) | 0.482 | 1 | 0.488 |
| Female | 131(100.0) | 63(48.1) | 68(51.9) | | | |
| **Total** | **641(100.0)** | **291(45.4)** | **350(54.6)** | | | |
| **Age (years)** | | | | | | |
| 15-19 | 18(100.0) | 6(33.3) | 12(23.4) | 2.321 | 9 | 0.004 |
| 20-24 | 150(100.0) | 63(42.0) | 87(24.9) | | | |
| 25-29 | 263(100.0) | 108(41.1) | 155(44.3) | | | |
| 30-34 | 137(100.0) | 80(58.4) | 57(16.3) | | | |
| 35-39 | 45(100.0) | 26(57.8) | 19(5.4) | | | |
| 40-44 | 18(100.0) | 4(22.2) | 14(4.0) | | | |
| 45-49 | 6(100.0) | 2(33.3) | 4(1.1) | | | |
| 50-54 | 2(100.0) | 0(0.0) | 2(0.6) | | | |
| 55-59 | 1(100.0) | 1(100.0) | 0(0.0) | | | |
| ≥60 | 1(100.0) | 1(100.0) | 0(0.0) | | | |
| **Total** | **641(100.0)** | **291(100.0)** | **350(100.0)** | | | |
| **HIV Status** | | | | | | |
| Positive | 162(100.0) | 75(46.3) | 87(53.7) | 2.321 | 2 | 0.313 |
| Negative | 423(100.0) | 196(46.3) | 227(53.7) | | | |
| Not Disclosed | 56(100.0) | 20(35.7) | 36(64.3) | | | |
| **Total** | **641(100.0)** | **291(45.4)** | **350(54.6)** | | | |
| **Population Type** | | | | | | |
| MSM | 500(100.0) | 224(44.8) | 276(55.2) | 3.074 | 2 | 0.079 |
| FSW | 115(100.0) | 51(44.3) | 64(55.7) | | | |
| GP | 26(100.0) | 16(61.5) | 10(38.5) | | | |
| **Total** | **641(100.0)** | **291(45.4)** | **350(54.6)** | | | |

systematically assessed the factors affecting completion of multi-dose vaccine schedules by adolescents [29]. With a p-value of 0.865, no association between gender and HBV vaccination completion rate was observed amongst children or younger populations in study conducted in Guinea [24]. However, this is in sharp contradiction to the findings of the study by Nnaemeka and Mbadiwe [19].

Despite women having more probability of initiating vaccination series, this study reveals a comparatively small gender gap in adherence and completion of HBV vaccination program. This suggests that comparable barriers to vaccine uptake, such as long waiting times at vaccination centers, transportation costs, misinformation about the vaccination program or lack of correct information, lack of adequate institutional supports, and economic problems, are common to both genders [7,25].

### Effect of HIV status on vaccination adherence and completion rate

Contrary to expectations, participants who lived with HIV had the same completion rate of 46.3% as the HIV-negative participants, while those with undisclosed HIV status had a completion rate of 35.7%. However, HIV-positive participants were 1.1 times more likely to adhere and complete the vaccination series than HIV negative participants (AOR: 1.1, 95% CI: 0.9, 1.3). Increased knowledge of the effect or consequences of HIV/HBV co-infection may have contributed to this

tendency, though not statistically significant ($\chi^2$ = 2.321, df = 2, p-value = 0.313). In line with results of this study, a previous study observed that there was no statistically significant relationship between HIV status and HBV vaccination completion rate [15]. On the contrary, other previous studies have reported that people living with HIV are exposed to unique challenges which can affect their ability to successfully adhere or complete HBV vaccination programs [20,29,30]. In addition, some studies have strongly attributed prevailing challenges suffered by HIV positive participants in terms of access to healthcare services and vaccination programs, to factors like stigma, financial barriers, and comorbidities [29–31]. Therefore, according to Adelekan et al., [30], proper integration of Hepatitis B vaccination into the routine HIV care and treatment programs across the nation, sending reminders about vaccination appointments, and improving access to lifesaving healthcare services, would significantly improve health services uptake, including HBV vaccination, amongst HIV positive individuals [30].

### Effect of 'population type' on vaccination adherence and completion rate

A cross-sectional study amongst MSM and Transgender Women (TGW) which was conducted in Lagos and Abuja in Nigeria, reported just 5% self-reported HBV vaccination coverage [15]. This is far lower than the 44.8% vaccination completion rate amongst the MSM in this study. MSM are slightly more likely to adhere, and complete HBV vaccination series compared to FSW and the general population (AOR: 1.2, 95% CI: 0.9, 1.5), but this result is not statistically significant (p-value = 0.079). MSM is a high-risk population and increased awareness of this reality can contribute to a higher likelihood of adherence to HBV vaccination schedules leading to higher vaccination completion rate, than other vulnerable populations [14,15].

## Conclusion

This study unveils the intricacy of attaining high vaccination completion rates among high-risk populations in Lagos State, Nigeria. Though gender, population types and HIV status did not show significant influence on HBV vaccination adherence and completion rate, it is noteworthy that men, and those who are HIV-negative were less likely to adhere and complete their vaccination schedules. This is attributable to suboptimal health-seeking behavior, nonchalant attitude or complacency towards uptake of healthcare services, especially in people who do not experience any pains or sickness. Giving that age is a strong factor that influences HBV vaccination adherence and completion rate, tailored interventions that address barriers faced by younger and older people, will tremendously improve HBV vaccination adherence and completion rates. Through such targeted programs, attaining behavioral change in men and HIV-negative people, public education, along with improved healthcare access, will help move Nigeria closer to achieving the WHO's vaccination targets and reducing the burden of Hepatitis B in high-risk populations.

### Implications and recommendations for public health policy

Based on the results from this study, the following recommendations will help to increase vaccine uptake and completion rates among high-risk populations in Lagos State.

1. Initiating targeted vaccination campaigns for younger and older populations through the various social media platforms and community outreach programs, will reduce the impact of age difference on the success of vaccination programs.

2. Institutionalizing the integration of Hepatitis B vaccination programs into the routine HIV care and treatment continuum, will increase access to and health service uptake

3. Establishing strong community engagement and education routine, will address common misconceptions about the Hepatitis B vaccine and strengthen the decision-making process the target population.

4. Exploring the integration of artificial intelligence (AI) tools and digital platforms to help create reminders for vaccination programs or other medication regimen.

## Limitations and future research

Despite the valuable findings from this study, the following limitations have been observed. First, as a retrospective study, relying on secondary data may not fully capture all relevant sociodemographic and cultural factors which can impact the success of vaccination programs. Also, data was collected only from the OSS Clinic in Lagos State and may not be generalizable to other regions in Nigeria or sub-Saharan Africa, consequently. A future prospective study that explores other socio-economic factors like income, educational level, and health literacy, would help to provide a more holistic view of the barriers to vaccine adherence and completion rates. We also recommend a post-vaccination antibody titre (anti-HBs) assessment in future studies, to evaluate immunological response and sero-protection in addition to vaccination completion.

## Acknowledgments

The authors appreciate the contributions of the data clerks, the program officers, the nurses, the doctors, the pharmacists and management of the OSS Clinic, where this research took place.

## Author contributions

**Conceptualization:** Elizabeth Shoyemi, Ojore Godday Aghedo, Collins Obi.

**Data curation:** Elizabeth Shoyemi, Ojore Godday Aghedo, Collins Obi, Chukwuebuka Peter Kelvin Uwaoma, Oluwafisayomi S. Azeez, Okikiola Oyeniyi.

**Formal analysis:** Ojore Godday Aghedo, Collins Obi.

**Funding acquisition:** Elizabeth Shoyemi.

**Investigation:** Elizabeth Shoyemi, Ojore Godday Aghedo, Chukwuebuka Peter Kelvin Uwaoma.

**Methodology:** Elizabeth Shoyemi, Ojore Godday Aghedo, Collins Obi.

**Project administration:** Elizabeth Shoyemi, Collins Obi, Chukwuebuka Peter Kelvin Uwaoma, Oluwafisayomi S. Azeez, Okikiola Oyeniyi.

**Resources:** Elizabeth Shoyemi, Collins Obi, Chukwuebuka Peter Kelvin Uwaoma, Oluwafisayomi S. Azeez.

**Software:** Ojore Godday Aghedo, Collins Obi.

**Supervision:** Elizabeth Shoyemi, Ojore Godday Aghedo, Collins Obi.

**Validation:** Elizabeth Shoyemi, Ojore Godday Aghedo, Collins Obi, Chukwuebuka Peter Kelvin Uwaoma, Oluwafisayomi S. Azeez, Okikiola Oyeniyi.

**Visualization:** Ojore Godday Aghedo, Collins Obi.

**Writing – original draft:** Ojore Godday Aghedo.

**Writing – review & editing:** Elizabeth Shoyemi, Ojore Godday Aghedo, Collins Obi, Chukwuebuka Peter Kelvin Uwaoma, Oluwafisayomi S. Azeez, Okikiola Oyeniyi.

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
