## [Decision Letter · Decision Letter 0]

27 Nov 2025

Dear Dr. Aghedo,

We look forward to receiving your revised manuscript.

Kind regards,

Jason T. Blackard, PhD

Academic Editor

PLOS ONE

Journal Requirements:

2. Thank you for stating the following in your manuscript:

[The HBV vaccination program that gave rise to the data used in this research was funded by the U.S. Department of Defense (DoD), through the Henry M. Jackson Foundation Medical Research International (HJF-MRI). However, the findings and opinions expressed in this article do not represent opinions of DoD or HJF-MRI, they are solely those of the authors. No specific external funding was received for this research. ]

[The author(s) received no specific funding for this work.]

3. In the online submission form, you indicated that [We have not made the data used in this research publicly available yet, to protect the participants who are high-risk populations from possible breach of confidentiality, and stigma associated with same-sex behavior and commercial sex work. However, the data are available at the OSS clinic of CPHI, upon reasonable request it will be made available. All requests should be sent to info@cphinigeria.org or to the corresponding author. Also, no online supplemental information is required because all data collected has been analyzed and reported in this article.].

Additional Editor Comments:

This is a cross-sectional study of HBV vaccination rates conducted in Lagos, Nigeria.

The methods and results are described well, although the findings are not particularly surprising.

In addition to the comments raised by the two reviewers, additional clarifications below would strengthen the manuscript further:

The introduction must include information about the availability of HBV vaccination in the public and private sectors across Nigeria as well as any associated cost.  Similarly, what are the nationwide policies/guidelines on HBV vaccination?Are HBV medications available in the public and private sectors?

It is unclear why data prior to 2021 were not included.

Reviewers' comments:

Reviewer's Responses to Questions

**Comments to the Author**

1. Is the manuscript technically sound, and do the data support the conclusions?

Reviewer #1: Partly

Reviewer #2: Yes

2. Has the statistical analysis been performed appropriately and rigorously?

Reviewer #1: I Don't Know

Reviewer #2: Yes

3. Have the authors made all data underlying the findings in their manuscript fully available?

Reviewer #1: No

Reviewer #2: Yes

4. Is the manuscript presented in an intelligible fashion and written in standard English?

Reviewer #1: Yes

Reviewer #2: Yes

Reviewer #1: This is an automated report for PONE-D-25-10988. This report was solicited by the PLOS One editorial team and provided by ScreenIT.

ScreenIT is an independent group of scientists developing automated tools that analyze academic papers. A set of automated tools screened your submitted manuscript and provided the report below. Each tool was created by your academic colleagues with the goal of helping authors. The tools look for factors that are important for transparency, rigor and reproducibility, and we hope that the report might help you to improve reporting in your manuscript. Within the report you will find links to more information about the items that the tools check. These links include helpful papers, websites, or videos that explain why the item is important. While our screening tools aim to improve and maintain quality standards they may, on occasion, miss nuances specific to your study type or flag something incorrectly. Each tool has limitations that are described on the ScreenIT website. The tools screen the main file for the paper; they are not able to screen supplements stored in separate files. Please note that the Academic Editor had access to these comments while making a decision on your manuscript. The Academic Editor may ask that issues flagged in this report be addressed. If you would like to provide feedback on the ScreenIT tool, please email the team at ScreenIt@bih-charite.de. If you have questions or concerns about the review process, please contact the PLOS One office at plosone@plos.org.

Reviewer #2: The study investigated the impact of demographic factors and HIV status on HBV vaccination adherence

and completion rate among high-risk populations such as MSM and FSW in Lagos State, Nigeria. The study has been conducted and analysed the data appropriately. The study concluded that there are certain demographic factors playing a role in the uptake of HBV vaccinations among high-risk populations. The manuscript may be accepted for publication after making some minor changes as listed below:

1. It is assumed that the vaccination schedule followed in this study was 0-1-6 and it is important to include these details in the methodology section.

2. In line No.189, the age range mentioned is 5 - 29 years and this typo may be corrected.

3. In line 322, is it 'holistic' or 'wholistic'?

4. In futurology, maybe the authors could consider including antibody titre post vaccination, and this will assess the respondent rate.

**Do you want your identity to be public for this peer review?** For information about this choice, including consent withdrawal, please see our Privacy Policy

Reviewer #1: No

Reviewer #2: **Yes:** Pachamuthu Balakrishnan

---

## [Author Response · Author response to Decision Letter 1]

8 Jan 2026

We sincerely thank the academic editor and the reviewers for the constructive and insightful comments on our manuscript. We have carefully addressed all editorial and reviewer comments. We believe the revised manuscript has been substantially improved as a result, and both the tracked-changes version and an unmarked clean copy have been uploaded. Our point-by-point response to the journal requirements, additional editor’s comments and the reviewers’ comments are provided in a file named, "Responses to Reviewers" and uploaded too for your reference.

---

## [Decision Letter · Decision Letter 1]

1 Feb 2026

Impact of demographic factors and HIV status on Hepatitis B vaccination adherence and completion rate among high-risk populations in Lagos State, Nigeria

PONE-D-25-10988R1

Dear Dr. Aghedo,

We’re pleased to inform you that your manuscript has been judged scientifically suitable for publication and will be formally accepted for publication once it meets all outstanding technical requirements.

Kind regards,

Jason T. Blackard, PhD

Academic Editor

PLOS One

Additional Editor Comments (optional):

None

Reviewers' comments:

Reviewer's Responses to Questions

**Comments to the Author**

Reviewer #2: All comments have been addressed

2. Is the manuscript technically sound, and do the data support the conclusions?

Reviewer #2: Yes

3. Has the statistical analysis been performed appropriately and rigorously?

Reviewer #2: Yes

4. Have the authors made all data underlying the findings in their manuscript fully available?

Reviewer #2: (No Response)

5. Is the manuscript presented in an intelligible fashion and written in standard English?

Reviewer #2: Yes

Reviewer #2: All the responses to the queries raised have been addressed appropriately. However, some minor corrections may be needed as follows:

1. Line 153: "subjects" may be replaced with "study participants".

2. If available, the manufacturer details of the vaccine used may be included in the methodology section appropriately.

**Do you want your identity to be public for this peer review?** For information about this choice, including consent withdrawal, please see our Privacy Policy

Reviewer #2: **Yes:** Pachamuthu Balakrishnan

---

## [Editor Report · Acceptance letter]

PONE-D-25-10988R1

PLOS One

Dear Dr. Aghedo,

I'm pleased to inform you that your manuscript has been deemed suitable for publication in PLOS One. Congratulations! Your manuscript is now being handed over to our production team.

Kind regards,

on behalf of

Dr. Jason T. Blackard

Academic Editor

PLOS One